

# An investigation into the response of the soil ecological environment to tourist disturbance in Baligou

Xiaolong Chen[1,2], Fangyuan Cui[3], Cora Un.In Wong[1], Hongfeng Zhang[1] and Feiyang Wang[4]

[1] Faculty of Humanities and Social Sciences, Macao Polytechnic University, Macao, China
[2] Department of Management, Henan Institute of Technology, Xinxiang, Henan, China
[3] Department of Mechanical Engineering, Henan Institute of Technology, Xinxiang, Xinxiang, Henan, China
[4] School of Economics, Trade and Management, Xinjiang Institute of Technology, Aksu City, China

Corresponding author
Xiaolong Chen,
osakacool@hait.edu.cn

## ABSTRACT

The purpose of this study is to understand the response patterns of the soil ecological environment of the Macau Wetland Park to different levels of tourist interference and to provide a scientific basis for the rational development of the Bali Gou ecological tourism plan and the protection and management of the scenic area's ecological environment. Combine the methods of field collection and laboratory physical and chemical data analysis to analyze the impact of the strength of tourism disturbance on the soil ecological environment of Baligou. During the tourist activities in Baligou, the human factors in the process have an impact on the physical aspects of the scenic area's soil, such as soil bulk density, color tone, porosity, compactness, capacity, and leaf litter. In addition, pH value, soil enzyme activity, organic matter, and heavy metals in the soil have caused some interference. The overall data show that the dynamic of soil indicators is background area < buffer area < disturbed area, where the sample area is affected by human trampling and infrastructure construction and other disturbances leading to regular changes in the content of Pb and Cr in the sample soil, while the content of other heavy metals is lower than the national standard. The range of the most extreme values of the SRI integrated soil ecological response evaluation index was A1 disturbance area = 4.679 and C1 background area = 1.263, respectively. The larger the value of the SRI response index, the greater the impact and negative effect of the disturbance influence of tourism activities on the soil ecological environment of the scenic area, and the higher the level of response of the soil ecological environment of the scenic area to the disturbance of tourism activities. Moderate and reasonable tourism development activities play a role in promoting soil quality in scenic areas. Therefore, it is suggested to carry out reasonable ecological environment planning and take certain macro-measures to prevent the deterioration of the soil ecological environment, so as to achieve a win-win situation of ecological environmental protection while developing the tourism economy in scenic areas.

## INTRODUCTION

Tourism scenic area is an important carrier for people's leisure and entertainment for tourism activities (*Li, Pu & Zhang, 2012*), and the quality of ecological environment of tourism scenic area directly determines the determines its sustainability and the smooth and the smooth implementation of tourism planning. With the improvement of people's living standards, the number of trips to tourist attractions has increased by leaps and bounds. Human interference has caused certain pressure and damage to the carrying capacity of tourist scenic spots (*Gong, Lu & Jin, 2009*), and a series of irreversible soil environmental pollution problems such as slabbing, soil erosion and uneven distribution of trace elements in tourist scenic spots due to human trampling and other factors have become more and more obvious (*Hao, Zheng & Zhu, 2021*).

The Bailigou Scenic Spot is located in Hui County, Xinxiang City, in the Central Plains hinterland, and is an important tourist scenic area in northern Henan Province, which has an important strategic significance in the ecological construction pattern of high-level tourist scenic areas. As the Bailigou Scenic Area was selected as a national 5A tourist scenic area in December 2019, the peak number of visitors was significantly higher than in previous years, thus causing great pressure on the carrying capacity of the tourist scenic area. Human disturbances such as trampling by tourists will damage the vegetation and litter layers on the soil surface, resulting in a decrease in soil moisture content and loam organic matter content. These factors may lead to changes in soil pH. Secondly, the number of tourists in the Baligou Scenic Spot is larger than that of Jiulian Mountain and Tianjie Mountain, and the effect of tourists' trampling is significant. Building construction and alkaline building residues in the Baligou Scenic Spot are serious, which is also the reason for the high pH value in the disturbed area of the Baligou Scenic Spot. The edge of the tourist scenic area and the core play area, soil surface is affected by different degrees of human trampling (*Xiao et al., 2016*), the soil ecological environment has been greatly damaged, causing a certain resistance to the sustainable development of the ecological environment of the tourist scenic area, which will further cause a risk of reducing the high quality and high value of tourism products to the various natural tourism resources of the tourist scenic area (*Dai, Xia & Li, 2021*). Once the natural resources are damaged and polluted, the cost of further restoration and purification is too high.

The current research related to soil ecology in scenic areas has become a hot spot in regional tourism research. Domestic and international studies have shown that tourism disturbances have had far-reaching effects on soil organic matter (*Tang & Yang, 2014*; *Zhao & Hou, 2019*; *Pulido-Fernandez, Casado-Montilla & Carrillo-Hidalgo, 2019*), acidity and soil erosion (*Folgado-Fernandez, Campon-Cerro & Hernandez-Mogollon, 2019*; *Li et al., 2017*; *Yang & Dou, 2022*). However, these studies mainly focus on the measurement and comparison of soil physical and chemical properties in scenic areas, the research index factors are relatively single and general, the correlation study of the impact of different disturbance intensities and regional influences on the soil ecological environment is insufficient, and the study of the response of tourism disturbance on the soil ecological environment in central mountainous scenic areas is still lacking. The research on the

 

soil ecological environment of the South Taihang Mountains along the Yellow Basin, in particular, is still in the early stages.. Therefore, it is important to understand the response law of the soil ecological environment in Bailigou scenic area with respect to different tourism disturbance strengths and weaknesses, in order to provide a scientific basis for the rational development of Bailigou scenic area ecotourism planning and ecological environmental protection and management.

## MATERIALS AND METHODS

### Research area

The Baligou Scenic Spot is located in the northwest of Xinxiang City, Henan Province, in the center of the hinterland of northern Henan. The overall planning covers an area of 109 square kilometers. The Baligou Scenic Spot mainly includes three parts: Baligou Tourist Area, Tianjie Mountain Tourist Area and Jiulian Mountain Tourist Area. The scenic spot belongs to the warm temperate continental monsoon climate, which is affected by the trend of the South Taihang Mountains and the altitude of the scenic mountains, and the monsoon effect is obvious. The four seasons are distinct and the climate is pleasant. The average annual temperature is 12 °C ~14 °C. It is the first choice for tourists from all over the country to escape the summer heat. The annual precipitation is 576.6 mm, and the frost-free period is 214 days throughout the year. The annual average sunshine time is 2020.1 h. The soil in the scenic spot belongs to cinnamon soil and brown loam soil, and the forest coverage rate of Baligou Scenic Spot is as high as 90%. The soil profile layer is more obvious, and the soil texture is more sticky and heavier than that of other plain areas.

The Baligou Scenic Spot contains a variety of typical temperate deciduous broad-leaved trees that are bountiful in biological resources. The picturesque area combines key appealing characteristics, such as the essence of the South Taihang Scenic Area's sceneries, with tourist resources, such as lovely valleys, forest trails, health and leisure opportunities, and cultural heritage. It draws an increasing number of people as a well-known national 5A-level tourism destination. In 2019, more than 1.6 million people visited the magnificent area each year.

### Quadratic setup and soil sample collection

Previous studies have shown that tourists in the scenic spot have a great impact on the physical and chemical properties (*Li, Yang & Xiao, 2010*; *Lu, Gong & Jin, 2011*; *Duan & Zhu, 2019*; *Zhang, 2020*), and others have shown in their studies that tourists in the scenic spot have a great impact on the physical and chemical properties of the soil within 3 m of the extension of the scenic trail through trampling and other artificial disturbances.

During the sample selection process, the research team collected the soil around the tourist plank road in Baligou Scenic Spot and obtained the permission from the Xinxiang Nantaihang Tourism (Group) Co., Ltd. and the Baligou Tourism Scenic Area Committee. The researchers obtained samples in the Baligou Scenic Spot. Obtaining samples (soil, leaf thickness, *etc.*) is very common and will not cause damage to the environment. The sampling has been recognized by local residents and tourist attractions.

The physical and chemical properties of soil samples in the extension area did not change significantly according to the characteristics of appearance factors such as the geographic location of Baligou Scenic Spot, the direction of the trails, and the distribution space of core scenic spots, this study selected six soil sampling points (two in Baligou tourist area, two in Tianjieshan tourist area, and two in Jiulianshan tourist areas), for a total of 18 different points distributed in the scenic area. The selected sampling points are basically located near the main scenic spots and tourist service centers, and the sampling points of the Baligou tourist area are selected in the Tianhe Waterfall and the tourist service center sections. The sampling points of the Tianjie Mountain tourist area are selected in the Laoye top scenic section and the tourist service center section. The sampling points of the sample plot in the Jiulian Mountain tourist area are selected at the Xilian Temple in the Xiaoxi Tianjing Section and the central section of the Xilianxia Scenic Spot. The distribution map of sampling locations is shown in Fig. 1. The sample belt was set along the vertical direction of the scenic spot's main tourist trail, and the main tourist trail in each tourist area was selected with a distance of 0.12 m (the tourism activity interference area), 3–5 m (tourist activity buffer area), and 6–10 m (tourist activity back ground area) for a 1 m × 1 m quadrats. In each transect along the travel trail, in each selected sample quadrat of activity disturbance, according to the small quadrat with an area of 10 cm × 10 cm, soil sampling tools were used to collect the surface soil of 0-12 cm in the sample area in the target area, and take the sample. The amount of soil was 500 g per sample point, and samples were taken in polyethylene sample bags, and each sample was numbered. The soil samples were screened for stones and animal and plant residues, and a preliminary micro-treatment was carried out. The samples were sealed in the original soil and placed in the laboratory for natural air-drying, with subsequent regular analysis of the physical and chemical properties of the soil in the scenic spot.

## Determination of soil physical and chemical properties

By drying the soil at 110 °C, the natural soil moisture content was identified. The sample data were then organized and examined. The pH of the soil samples in the scenic area was measured by potentiometric method (the ratio of water-extracted soil liquid was 2.5:1); the soil firmness index was measured by a firmness meter, and the measurement was based on the GB 7843-1987 Determination Standard for Forest Soil Firmness; soil natural moisture content Measured by drying method at 110 °C, soil brightness was measured by comparison with Mensell soil color chart. The external heating $K_2Cr_2O_7$ bulk density ratio method was used to determine the ecological content of soil organic matter. Soil organic matter content was determined by potassium dichromate method (chemical oxygen demand was determined by reducing substances in oxidized samples). The content of heavy metals in soil was determined by ICP-MS inductively coupled plasma mass spectrometer. For specific detection methods, please refer to the references (Table 1) (*Ni, Peng & Gao, 2016*; *Deng, Zhang & Shi, 2018*; *Ma et al., 2016*).

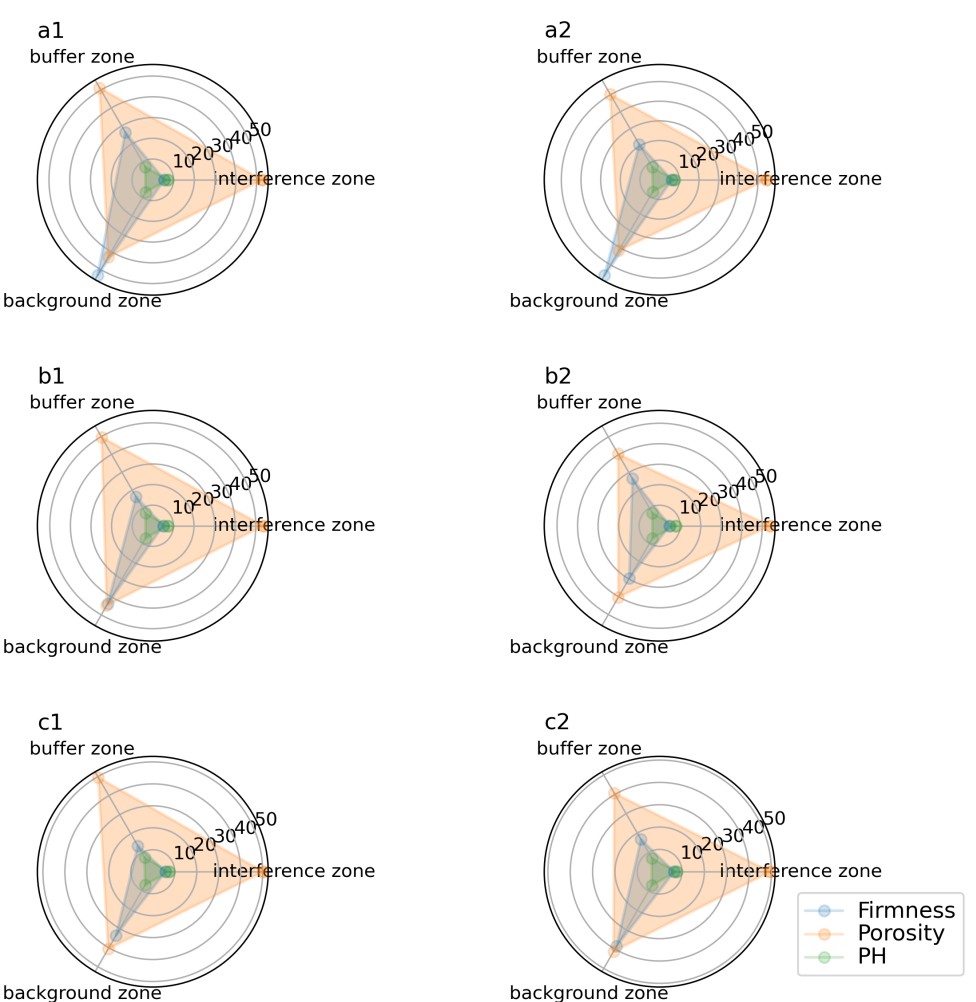

**Figure 1** Response data of soil compactness, porosity and soil PH value to tourism disturbance.

## Construction of soil ecological environment response evaluation model
### Construction of single factor response model
It is affected by tourists, disturbing factors, and the natural factors of the landscape itself.The influence factor of the increase and decrease changes was determined primarily through the sampling point soil collection analysis.

### Soil ecological environment comprehensive response evaluation index model
Tourism disturbance has a multifaceted impact on the soil ecological environment. This research considers the response evaluation of soil single factor based on the comprehensive consideration, combined with the processing and analysis of related data.According to the comprehensive evaluation model of soil environments proposed by *Liu, Qin & Xie (2006)*. In this way, the comprehensive response of different influencing factors to the soil ecological environment of the Baligou Scenic Spot under different tourism disturbance

**Table 1  Determination methods and parameters of soil physical and chemical properties.**

| Determination of samples | Method employed | Parameter | Citation |
|---|---|---|---|
| Soil sample pH | Potentiometric determination | The ratio of water leaching and extraction soil liquid is 2.5:1 | Ni et al. (2019) |
| Soil firmness | Firmness Meter Determination | GB 7843-1987 Determination Standard of Forest Soil Firmness | Ni et al. (2019) and *Ma et al. (2016)* |
| Soil natural moisture content | 110 °C drying method | >1 mm air-dried soil, moisture percentage based on air-dried soil | *Deng, Zhang & Shi (2018)* and *Ma et al. (2016)* |
| Soil brightness | Mensell soil color chart comparison method | Divided into equal parts from 0-10 from dark to light | *Deng, Zhang & Shi (2018)* and *Ma et al. (2016)* |
| Ecological content of soil organic matter | External heating K2Cr2O7 bulk density ratio method | National standard substances (GBW07412a and GBW07415a) are samples | *Ma et al. (2016)* |
| Soil organic matter content | Potassium dichromate method | National standard substances (GBW07412a and GBW07415a) are samples | *Ma et al. (2016)* |
| Soil heavy metal content | ICP-MS Inductively Coupled Plasma Mass Spectrometry Method | Determination of different isotopes of elements to determine metal content | Ni et al. (2019), *Deng, Zhang & Shi (2018)* and *Ma et al. (2016)* |

intensities was evaluated. The index evaluation model is expressed as:

$$\text{SRI} = \sum_{i=1}^{i-1} w_i \sum_{j=1}^{9} \frac{S_{ij} - S_{oj}}{S_{oj}}. \tag{1}$$

Among them, it represents the comprehensive response evaluation index of the soil ecological environment of the scenic spot; $j$ represents the number of indexes of influencing factors, taking 9 weight indicators such as soil bulk density, water content, porosity, soil compactness, pH value, soil pH, enzyme activity and organic matter. $S_{ij}$ is the measured value of the $j$th impact factor in the $i$th sample soil.

## Analysis and processing of experimental data

The SPSS analytic program was used to sort and analyze the sample data from the disturbance region, buffer area, and background area. Excel was used to perform data analysis and processing on sample data. First, a single-factor response model was constructed for soil samples in different zones, and then data analysis was performed on the soil ecological environment comprehensive response evaluation index model.

Using multivariate statistical factor analysis and a comprehensive response evaluation index analysis method, determine the weight of each index factor, and measure soil quality.The correlation test of statistical data was completed using the data statistical software *SPSS 19.0*, and the visualization maps such as the study area were drawn with *ArcGIS 10.5* version.

## RESULTS AND ANALYSIS

### The influence of tourism disturbance on soil physical properties
*Response of leaf litter and soil brightness to tourism disturbance*

In the selected 6–10 m background area of tourism activities in the scenic transect, through the observation of the soil surface and vertical section of the background area, it can be seen that the vegetation on the soil surface is well protected, and there are generally residual litter on the soil surface. The thickness of the obtained data generally ranges from 0 to four cm (Table 2). There is a humus layer of about two cm in some samples in the outer edge area. The litter layer is sensitive to tourism disturbance. The reason may be that the litter layer is attached to the soil surface and has a loose structure. Environmental conditions such as tourism trampling, manual cleaning and even strong wind will affect the accumulation and decomposition of the litter layer process.

In areas where there is frequent tourist activity, soil erosion is more evident due to the trampling of tourists and the accumulation of tourist garbage. This disturbance can cause the soil surface to become exposed, leading to rocky desertification in some cases. As a result, vegetation on the soil surface and within the soil profile can become scarce. The reduction of vegetation and tree leaves leads to a decrease in the decayed rot layer of animals and plants. This reduction in organic matter content of the soil components can result in an increase in soil brightness and chroma. In areas of tourist activity, vegetation and litter become less visible, and the root system can also be significantly reduced compared to non-disturbed areas. Ultimately, the decrease in vegetation and litter can cause a reduction in soil organic matter content, leading to an increase in soil brightness and chroma.

According to soil color is one of the most important characteristics to observe when observing soil morphological characteristics. According to soil color and cross-sectional shape, the change in soil performance can be detected. and the change law of soil performance is detected according to soil color and cross-sectional shape. The soil brightness was compared using the Mensell soil color chart, and it was found that the color of the surface soil in the sample area of Baligou in the wet state was as follows: brown (7.5YR4/3), but the buffer zone and background area were relatively less affected by human interference, and the soil color was changed from brown to dark brown (7.5YR3/3) and dark brown (7.5YR2 /2) after color card comparison data (Table 2).

*The response of soil bulk density to the disturbance of tourism trampling*

The findings indicated that soil bulk density increased in a favorable direction as tourist disturbance intensity increased. The soil bulk density exhibits a decreasing tendency as one travels more and farther from the touristy region. Table 3 displays the results of the collection and examination of the particular soil samples.

*The response of soil water content to the disturbance of tourism trampling*

It can be seen from Table 3 that the data analysis and sorting through the analysis software SPSS shows that the soil moisture content in the disturbance area with large tourist activity is the lowest (30%), followed by the buffer zone, and the background area has the highest water content (67%).

**Table 2  Response data of litter and soil brightness to tourism disturbance.**

| Sample Area | Sample zone | Sample | Thickness of fallen leaves/cm | Soil color |
|---|---|---|---|---|
| Baligou Sample Area A1 | Interference zone | A1 | 0 | Brown (*7.5YR4/3*) |
| | Buffer zone | A2 | 1 | Dark brown (*7.5YR3/3*) |
| | Background zone | A3 | 2 | Dark brown (*7.5YR2/2*) |
| Baligou Sample Area A2 | interference zone | A4 | 0 | Dark brown (*7.5YR3/3*) |
| | Buffer zone | A5 | 1 | Old dark brown (*7.5YR3/1*) |
| | Background zone | A6 | 3 | Black (*7.5YR2/1*) |
| Tianjieshan sample Area B1 | Interference zone | B1 | 0 | Reddish brown (*2.5YR4/8*) |
| | Buffer zone | B2 | 2 | Reddish brown (*2.5YR3/4*) |
| | Background zone | B3 | 4 | Very dark russet (*2.5YR2/2*) |
| Tianjieshan sample Area B2 | Interference zone | B4 | 0 | Dark reddish brown (*5YR3/4*) |
| | Buffer zone | B5 | 1 | Very dark russet (*5YR3/3*) |
| | Background zone | B6 | 3 | Dark brown (*5YR2/2*) |
| Jiulianshan sample Area C1 | Interference zone | C1 | 0 | Reddish brown (*5YR4/4*) |
| | Buffer zone | C2 | 1 | Dark reddish brown (*5YR3/3*) |
| | Background zone | C3 | 3 | Dark brown (*5YR2/1*) |
| Jiulianshan sample Area C2 | Interference zone | C4 | 0 | Reddish brown (*5YR4/4*) |
| | Buffer zone | C5 | 2 | Reddish brown (*5YR3/3*) |
| | Background zone | C6 | 3 | Reddish brown (*2.5YR3/4*) |

**Table 3  Response data of soil moisture content to trampling interference.**

| Sample area | Sample zone | Sample | Soil moisture content | Soil bulk density (g cm$^{-3}$) |
|---|---|---|---|---|
| Baligou Sample Area A1 | Interference zone | A1 | 30% | 129.6 |
| | Buffer zone | A2 | 37% | 108.4 |
| | Background zone | A3 | 62% | 101.3 |
| Baligou Sample Area A2 | Interference zone | A4 | 32% | 119.6 |
| | Buffer zone | A5 | 38% | 98.4 |
| | Background zone | A6 | 67% | 91.2 |
| Tianjieshan sample Area B1 | Interference zone | B1 | 32% | 132.5 |
| | Buffer zone | B2 | 41% | 116.1 |
| | Backgroundzone | B3 | 62% | 100.3 |
| Tianjieshan sample Area B2 | Interference zone | B4 | 30% | 131.6 |
| | Buffer zone | B5 | 43% | 109.3 |
| | Background zone | B6 | 66% | 100.6 |
| Jiulianshan sample Area C1 | Interference zone | C1 | 31% | 132.3 |
| | Buffer zone | C2 | 46% | 123.6 |
| | Background zone | C3 | 63% | 103.3 |
| Jiulianshan sample Area C2 | Interference zone | C4 | 32% | 136.6 |
| | Buffer zone | C5 | 43% | 115.4 |
| | Background zone | C6 | 65% | 102.6 |

### Response of soil porosity to trampling disturbance

One of the key metrics for assessing soil structure is soil porosity. The looseness of the soil in the picturesque location and the greatest amount of air and water that may be accommodated are both determined by the porosity of the soil. If the soil porosity is smaller, the soil capacity is smaller, and the soil is compact and not loose. On the contrary, the opposite is true. By comparing the samples in the interference area, buffer area and background area, it is found that the soil porosity in the background area is much larger than that in the interference area and buffer area (Fig. 1). It shows that the background area with long distance interval has larger soil capacity, loose soil and good soil structure.

### Response of soil compaction to trampling disturbance

One of the key components of the depiction of soil attributes is soil compaction, commonly referred to as soil hardness or soil penetration resistance. Sampling and data reading of the soil in the sample region revealed that the interference area, buffer zone, and background area of soil compaction were all present (Fig. 1). The soil hardness in the disturbance zone is much greater than that in the background zone and buffer zone. The data show that in the background area spaced far away, the soil hardness is the least, the soil is loose, and the soil structure is in good condition. The soil hardness in the interference area with a large value is relatively large, and compacting the soil will prevent the infiltration of water, thereby reducing the absorption and utilization rate of soil water and fertility, and further affecting the growth of plant roots in the scenic area.

## The influence of tourism disturbance on soil chemical properties
### Response of soil pH to tourism disturbance

The sampling results of the sample area showed that the soil pH value was the highest in the disturbance area with dense tourists. With the increase of the edge distance, the pH value of the soil chemical properties in the sample area decreased due to the reduction of human interference factors such as tourists trampling on the background area (Fig. 1). The results of multiple comparisons showed that the pH of soil samples was in the order of tourism interference area >buffer zone >background area.

Data collection and analysis of sample regions in various parts of the nice location reveal that soil enzyme activity responds more visibly to tourist disturbance (Fig. 2). The invertase and catalase activities in the soil demonstrated that the interference region was noticeably larger than the background area and that the interference area was greater than the buffer area and the background area. The exact reverse is true in terms of urease activity performance, suggesting that the interference region, buffer area, and background area are all less than the long-distance background area. Human interference had less impact on the alkaline phosphatase activity, and the results showed no significant changes.

### Response of soil organic matter to tourism disturbance

Organic matter is an important factor affecting soil fertility. The proportion of organic matter in soil is directly related to soil properties and surface vegetation nutrition. The treading and picking activities of tourism activities cause serious damage to the litter layer and humus layer of trees in forest areas, which directly leads to the relative reduction

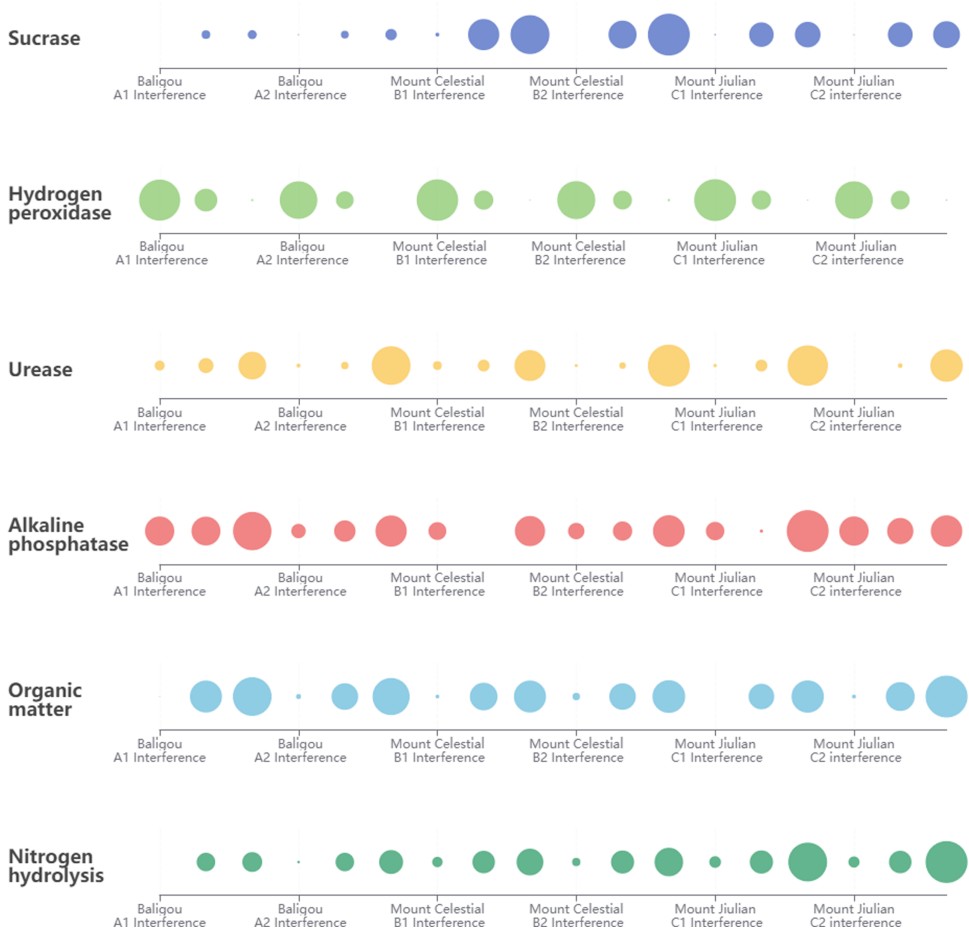

**Figure 2** Response data of soil enzyme activities to tourism disturbance.

of the amount of plants returned. At the same time, the compactness of soil compacted by tourism activities such as trampling directly affects the growth and development of plant roots, which will cause the decrease of organic matter content. In addition, the change of soil physical and chemical properties will reduce the number of some animals and microorganisms beneficial to the soil, and also lead to the reduction of soil organic matter content in the local soil. With the increase of the distance from the walking path in the sampling zone, the content of organic matter in the soil increased. The results of multiple sample comparisons of the six sample areas showed that the disturbance area was significantly affected, while the variation range of data in the background area and buffer zone was small (Table 4). Human disturbance such as tourist trampling reduces the content of organic matter in the soil.

### Response of soil heavy metal content to tourism disturbance

The conclusions of the extraction and comparison of the soil's heavy metal content in the sample area demonstrate that the levels of the heavy metallic components Hg, Cd, As, Pb, and Cr are all low and that the comparative content is below the national standard limit

**Table 4 Response of soil hydrolyzed nitrogen and organic matter to tourism disturbance.**

| Sample area | Sample zone | Sample | Organic matter/g kg$^{-1}$ | Hydrolysis nitrogen/mg kg$^{-1}$ |
|---|---|---|---|---|
| Baligou Sample Area A1 | Interference zone | A1 | 2.017 | 56.83 |
| | Buffer zone | A2 | 4.961 | 108.49 |
| | Background zone | A3 | 5.581 | 112.30 |
| Baligou Sample Area A2 | Interference zone | A4 | 2.435 | 63.52 |
| | Buffer zone | A5 | 4.463 | 108.56 |
| | Background zone | A6 | 5.437 | 123.6 |
| Tianjieshan sample Area B1 | Interference zone | B1 | 2.325 | 85.81 |
| | Buffer zone | B2 | 4.552 | 119.63 |
| | Backgroundzone | B3 | 4.962 | 131.64 |
| Tianjieshan sample Area B2 | Interference zone | B4 | 2.667 | 79.63 |
| | Buffer zone | B5 | 4.451 | 121.30 |
| | Background zone | B6 | 5.029 | 136.52 |
| Jiulianshan sample Area C1 | Interference zone | C1 | 1.984 | 89.36 |
| | Buffer zone | C2 | 4.361 | 121.68 |
| | Background zone | C3 | 4.989 | 165.32 |
| Jiulianshan sample Area C2 | Interference zone | C4 | 2.335 | 87.92 |
| | Buffer zone | C5 | 4.667 | 119.67 |
| | Background zone | C6 | 5.883 | 173.82 |

in the three tourism attractions.However, due to the influence of human trampling and infrastructure construction in the sample area, the content of Pb and Cr in the sample soil showed a regular trend. The data in Table 5 shows that the highest value of Pb interference area is 29.434 mg kg$^{-1}$, and the minimum value is 20.016/mg kg$^{-1}$.

The overall performance is interference area>buffer area>background area, and its regular performance is that the concentration of the heavy metals Pb and Cr in the soil increases with proximity to the main trail, as well as with the level of human involvement.

## The Evaluation of soil ecological environment comprehensive response evaluation index model

In Baligou sample A1 area and Jiulianshan sample C1 region, respectively, the highest range of the SRI Thorough Response Evaluation Index of Soil Ecological Environment was SRI = 4.679 and SRI = 1.263. In associated with data analysis, the higher the value of the SRI response index, the greater the impact and terrible effect of the disturbance of tourism activities on the soil ecological environment of the scenic area, and the higher the disturbance response level of the scenic area's soil ecological environment to tourism activities. On the other hand, there is less of an effect of tourism activities on the physiological environment of the soil.

As can be shown, the quantity to which tourist activity interference affects the Baligou Scenic Spot's ecological environment and the degree to which soil ecological environment interference influence tourism activity range from high to low:interference zone (A1 >B1 >A2 >B2 >C2 >C1), buffer zone (A1 >A2 >B1 >C1 >B2 >C2), background zone(A1 >A2

**Table 5  Response data of soil heavy metal content to trampling interference.**

| Sample area | Heavy metal | Heavy metal content/mg kg$^{-1}$ | | |
|---|---|---|---|---|
| | | Interference zone | Buffer zone | Background zone |
| Baligou Sample Area A1 | Hg | 0.026 | 0.028 | 0.031 |
| | Cd | 0.103 | 0.071 | 0.067 |
| | Pb | 1.724 | 1.257 | 1.126 |
| | Cr | 27.636 | 27.109 | 23.124 |
| Baligou Sample Area A2 | Hg | 0.031 | 0.041 | 0.033 |
| | Cd | 0.098 | 0.052 | 0.085 |
| | Pb | 3.426 | 2.138 | 1.438 |
| | Cr | 25.711 | 23.257 | 20.016 |
| Tianjieshan sample Area B1 | Hg | 0.011 | 0.009 | 0.025 |
| | Cd | 0.099 | 0.091 | 0.021 |
| | Pb | 5.483 | 4.134 | 2.371 |
| | Cr | 20.654 | 18.015 | 17.108 |
| Tianjieshan sample Area B2 | Hg | 0.038 | 0.019 | 0.014 |
| | Cd | 0.212 | 0.023 | 0.021 |
| | Pb | 4.065 | 3.216 | 1.072 |
| | Cr | 29.439 | 27.003 | 22.064 |
| Jiulianshan sample Area C1 | Hg | 0.020 | 0.012 | 0.023 |
| | Cd | 0.101 | 0.036 | 0.014 |
| | Pb | 4.490 | 3.002 | 1.082 |
| | Cr | 28.331 | 22.146 | 23.017 |
| Jiulianshan sample Area C2 | Hg | 0.019 | 0.023 | 0.012 |
| | Cd | 0.106 | 0.039 | 0.016 |
| | Pb | 4.382 | 3.165 | 1.094 |
| | Cr | 29.430 | 22.136 | 22.037 |

>B1 >C1 >B2 >C2) (Table 6). The highest SRI data was found in the disturbed area, and the lowest was in the background area. The SRI data of the soil ecological environment comprehensive response evaluation index in the disturbed area was higher than that in the background area. It indicates that the greater the impact and negative effect of tourism activities on soil ecological environment, the higher the response level of soil ecological environment to tourism activities.

## DISCUSSION

### The response of soil physical and chemical indicators to tourism disturbance

Through the data collection and analysis of the sample soil, the research results show that in the tourist activity interference area with a large flow of people, as the number of tourists increases, the interference intensity on the scenic spot soil increases, and the soil pH value shows a trend of gradual increase. The results of multiple comparisons showed that the soil sample pH interference area >buffer area >background area. The main reason is that

**Table 6 SRI data of comprehensive response evaluation index of soil ecological environment.**

| Sample area | SRI data of comprehensive response evaluation index of soil ecological environment | | |
|---|---|---|---|
| | Interference zone | Buffer zone | Background zone |
| Baligou Sample Area A1 | 4.679 | 3.106 | 2.031 |
| Baligou Sample Area A2 | 4.016 | 2.884 | 1.869 |
| Tianjieshan sample Area B1 | 4.362 | 2.039 | 1.866 |
| Tianjieshan sample Area B2 | 3.356 | 1.986 | 1.452 |
| Jiulianshan sample Area C1 | 3.296 | 2.013 | 1.751 |
| Jiulianshan sample Area C2 | 3.433 | 1.897 | 1.263 |

human disturbance such as trampling by tourists will damage the vegetation on the soil surface and the litter layer, which will lead to a decrease in the number of plants on the soil surface, serious soil compaction, resulting in soil erosion, poor water permeability of the soil, and soil alkalinity Enhanced pH is relatively elevated.

The increase in tourist disturbance behaviors reduces soil aeration in the scenic area and increases soil compaction. Human disturbance exposed the soil surface in some trampled areas, and the petrochemical phenomenon was obvious. Dead roots, fallen leaves, and human trampling resulted in the reduction of plant roots and the deterioration of the humus layer, which eventually led to the soil organic matter content in the soil disturbance area of the scenic spot being much lower than that in the background area.

With the increase in distance from the walking trail to the sampling belt, the content of organic matter in the soil increased. The comparison results of multiple samples in the 6 sample areas showed that the interference area was significantly affected, and the background area and buffer zone data had a small change. Tourist human disturbances, such as stampedes of tourists, have reduced the organic matter content in the soil of scenic spots.

With the increase in tourism disturbance behavior, the soil aeration of the scenic area is weakened, and the compactness of the soil is increased. The most direct impact is the reduction of the decomposition speed of the soil due to animal and plant residues, resulting in a reduction of the output speed of the transformation of soil organic matter. Human interference has resulted in bare leakage of the soil surface in some trampled areas and obvious petrochemical phenomena. Withered roots and leaves, as well as human trampling, have caused plant roots to reduce the bad humus layer, and eventually the soil organic matter content in the scenic spot's soil disturbance area is far lower than in the background area.

The contents of Hg, Cd, As, Pb and Cr in the soil of the sample area are relatively small, and the comparative contents are all lower than the national standard limit. However, due to the influence of human trampling and infrastructure construction, the Pb and Cr content in the sample soil showed a regular trend, the overall performance was interference area >buffer zone >background area, and its regular performance was the distance from

the main trail. The more recent, the higher the intensity of human interference, the higher the corresponding content of soil heavy metals Pb and Cr.

Combined with data analysis, the impact intensity of tourism activities on the ecological environment of the Baligou Scenic Spot, the disturbance response level of the soil ecological environment in the scenic area to tourism activities from high to low is: interference zone (A1 >B1 >A2 >B2 >C2 >C1), buffer zone (A1 >A2 >B1 >C1 >B2 >C2), background zone (A1 >A2 >B1 >C1 >B2 >C2). It demonstrates that the soil ecological environment responds more strongly to tourist activities the greater the influence and impact of those activities on the environment. On the other hand, the level of the ecological environment's response to tourist activities decreases as the influence of tourism activities on the environment is reduced.

## The analysis of strategies to reduce the response of scenic soil ecological environment to tourism disturbance

### Combined with the integration model of tourism + ecological disciplines, the internal and external adjustment of ecological scenic spots

At the time node of long and short vacations, tourists in the scenic spot tend to spout during the holidays, and then the interference of human factors causes pressure and damage to the natural ecological environment of the scenic spot (*Guo et al., 2019*). *Zhu, Wang & Sun (2021)* and *Zhang & Zhao (2016)* pointed out in the relevant literature that in the peak tourist season, tourists should be guided and separated in time and space, and the number of tourists during peak periods should be strictly controlled to avoid excessive interference by tourists and exceed the self-purification of the soil ecosystem in the scenic area. ability to cause irreversible damage to the soil ecological environment. In terms of internal factors, the interference effect of ecology on the carrying capacity of the ecological environment should be actively explored, and the physical and chemical properties of the scenic soil should be tested experimentally according to the integration of tourism, soil science, and ecology to further determine its response to the soil ecological environment. It provides a scientific basis for the rational development of scenic eco-tourism planning and the ecological environment's protection and management of scenic spots.

### Reasonably plan the ecological layout of scenic spots and strengthen spatiotemporal monitoring of soil environment

On the basis of the original distribution of scenic spots in the scenic spot, rationally plan the layout of its functional divisions, and fully consider the habits and diffusion characteristics of tourists. Through the analysis of tourists' space routes, functional areas such as scenic trails, sightseeing bridges and rest areas can be reasonably planned. For example, the construction of "return" type tourist loops and gridded wide tourist trails can effectively divert and evacuate crowded people, and avoid excessive concentration of tourists and excessive trampling on the soil in local areas of the scenic spot, resulting in strong negative interference with soil performance. At the same time, it advocates the concept of green ecology and environmental protection, compares the materials of the trails, lays durable wooden boards, pebbles, and bionic wood and stone materials, and sets up extended

railings to reduce the ecological damage to the extended area of the trail caused by travel interference.

The main reason for the decline of soil environmental quality in scenic spots caused by tourists' human disturbance is external factors such as human trampling (*Hu & Huang, 2016*). The passenger flow is monitored in real time. If the passenger flow is large, the carrying capacity of the soil environment may be insufficient, and the passengers will be evacuated in time. While monitoring the passenger flow, various index factors (soil compactness, porosity, pH, litter layer thickness, *etc.*) of the soil in the scenic area are monitored, and remedial response measures are taken in time for changes in soil environmental factors.

### Strengthen the publicity and cultivation of environmental awareness and increase public participation

While experiencing the physical and mental enjoyment brought by the natural environment of the scenic spot, tourists strengthen their awareness of ecological environmental protection (*Xiu, Chen & Huang, 2019*). Encourage residents around the scenic area to feel the ecological and economic benefits brought about by protecting the ecological environment by participating in the development and management of natural ecological resources in the scenic area. Through the words and deeds of the scenic spot managers and destination residents, the surrounding propaganda and education are presented to the tourists, so as to improve the tourists' consciousness of protecting the natural ecological environment of the scenic spot, and advocate the instillation of green tourism, eco-tourism and low-carbon tourism awareness, so as to reduce the risk of excessive disturbance and destruction of scenic spots.

### Introduce ecological improvement governance system to improve soil ecological balance

Taking into account the negative impact of tourists' trampling and other factors on the soil in the scenic area, especially in the case of tourist disturbance areas, soil ecological improvement techniques in forest areas, ecological farmland or paddy fields should be introduced. At present, soil improvement technologies and methods are relatively concentrated in farmland irrigation and water conservancy improvement technology, biological improvement technology to prevent soil erosion by planting green manure, and chemical improvement technology to interfere with soil pH (*Liu, Li and Xu, 2020*; *Zhu, Ren and Tang, 2020*).. The scenic spot's soil ecology is systematically managed through soil improvement techniques and methods in order to achieve soil ecological balance.

## FUTURE RESEARCH PERSPECTIVES

Human factors in the process of tourism activities in the Baligou Scenic Spot, such as the proportion of soil, color, porosity, compactness, capacity, and deciduous layer, as well as the pH value, soil enzyme activity, organic matter, and soil heavy metals, have caused certain interference on the physical aspects of the area. However, this study has certain limitations. Due to the local nature of data collection, only conventional pollution

(human trampling, excessive water use, *etc.*) and conventional pollutants (particulate matter, sulfur oxide, carbon oxide, *etc.*) from human activities are considered in the factors affecting the soil ecological environment in the scenic area. Emerging pollutants refer to those substances that have emerged and gradually increased in recent years and are harmful to the environment and human health, such as chemicals containing antibiotics and pesticides, toxic metals in electronic waste, and synthetic micro-plastics. The impact of emerging pollutants on soil is very serious, because they enter the soil in different ways and pose a great threat to the soil ecosystem. For example, residues of antibiotics and pesticides can kill microorganisms in soil and cause damage to soil fertility; and toxic metals in e-waste can have toxic effects on organisms in soil. The synthetic microplastics can remain in the soil for a long time and affect the ecological balance of the soil.

The presence of manufactured nanomaterials in packaged food brought by tourists to scenic spots and their exposure to the environment, pharmaceutical and personal care products, nano and microplastics used in masks, food packaging materials, *etc.* All of them can cause important contamination of the soil ecology of the scenic area. Therefore, emerging contaminants have the potential to contaminate the ecology of any tourist attraction.

Nowadays, the study of emerging contaminants is a hot topic for future research.Therefore, when relevant data becomes available in the future, it has been effectively suggested to calculate emerging contaminant data as factors affecting soil ecological changes to obtain more accurate analysis results.

## CONCLUSIONS

In 2019, Baligou Scenic Spot was rated as a national 5A-level scenic spot through the quality rating of the General Administration of Quality Supervision, Inspection and Quarantine of the People's Republic of China, and its popularity has grown. Studies have shown that the Baligou Scenic Spot has caused certain disturbance and damage to the bulk density, compactness, pH value, moisture content, and chemical organic matter content of the soil in the scenic area due to human interference factors such as tourists' trampling, picking and basic construction of tourist attractions. Therefore, it is recommended to carry out reasonable ecological planning for the scenic spot, and take certain macro-solution measures to prevent the retrograde deterioration of the soil ecological environment, so as to truly realize the win-win situation of ecological and environmental protection while developing the tourism economy in the scenic spot.

### Funding

This research was funded by the National Natural Science Foundation of China, grant number 71864035, the Key Scientific Research Project of Colleges and Universities of Henan Province, grant number 19A790008, and the Doctoral Research Foundation of

Henan Institute of Technology (Project No: KQ2009). The funders had no role in study design, data collection and analysis, decision to publish, or preparation of the manuscript.

## Grant Disclosures

The following grant information was disclosed by the authors:

National Natural Science Foundation of China: 71864035.

Key Scientific Research Project of Colleges and Universities of Henan Province: 19A790008.

Doctoral Research Foundation of Henan Institute of Technology: KQ2009.

## Competing Interests

The authors declare there are no competing interests.

## Author Contributions

- Xiaolong Chen conceived and designed the experiments, performed the experiments, analyzed the data, prepared figures and/or tables, authored or reviewed drafts of the article, and approved the final draft.
- Fangyuan Cui analyzed the data, prepared figures and/or tables, and approved the final draft.
- Cora Un.In Wong conceived and designed the experiments, prepared figures and/or tables, authored or reviewed drafts of the article, and approved the final draft.
- Hongfeng Zhang conceived and designed the experiments, authored or reviewed drafts of the article, and approved the final draft.
- Feiyang Wang analyzed the data, prepared figures and/or tables, and approved the final draft.

## Data Availability

The raw data are available in the Supplemental Files.

## Supplemental Information

Supplemental information for this article can be found online at http://dx.doi.org/10.7717/peerj.15780#supplemental-information.

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
