# Peer review of "An investigation into the response of the soil ecological environment to tourist disturbance in Baligou"

_PeerJ, doi:10.7717/peerj.15780_

## Round 0.1 · original submission · Major Revisions

The article needs extensive language editing. The methodology part, especially the statistical analysis needs to be elaborated more.

Reviewer 1 ·

Basic reporting

2. INTRODUCTION Specified well about tourism promotion with scenic development and planning of Bailigou city (Guangdong Province) as in the accordance with soil ecological and environmental protection.
3. RESEARCH AREA Appreciable and wonder to see many heterogeneous factors which captivated and favoured the selected sample size area of research
4. MATERIALS & METHODS Appropriate and constructive with first hand info on analysis of physical and chemical properties of given soil with standard protocols and procedures


5. RESULTS AND ANALYSIS Outcomes of research are strongly qualified with proper insights into each and every mentioned parameter
Data Analysis and graphs are engaging into detailing of results
6. DISCUSSIONS Well articulated and very good at maintaining specificity in this kind of complex topic to understand the discussions with a straightforward portrayed explanations.
7. CONCLUSION Summarized very well about main finding of scenic spot with specific significance on aim of promoting agricultural tour as in connection with soil health and ecological protection.

Experimental design

Experimental design was aptly fitted to research work in a impressive way.

Validity of the findings

Outcomes of research are strongly qualified with proper insights into each and every mentioned parameter
Data Analysis and graphs are engaging into detailing of results
Well articulated and very good at maintaining specificity in this kind of complex topic to understand the discussions with a straightforward portrayed explanations.
Summarized very well about main finding of scenic spot with specific significance on aim of promoting agricultural tour as in connection with soil health and ecological protection.

Additional comments

Positive comments of Reviewer report :
• I recommend that this manuscript very strongly. The methodology, conceptual frame work of represented research work and data analysis are drawn in fascinated way with crystal clear explanations.
• I think this paper potentially added promise to new arena of research on China’s Agricultural tourism with novelity .
• Yes, I would recommend this paper with no corrections as above said. It is innovative, interesting and understand to read.
• I agree and recommend accepting this manuscript for publication because overall quality of this study is newer one and emerging concept. There are very few studies on this topic, more research is needed in this area for upgrading cultural heritage of different scenic spots in the world.
• Yes, I again believe this study will take a new path in a positive direction for soil conservationists and eco-tourism promoters.

·

Basic reporting

• Clear and professional English language is not used. Though I have corrected some typos and few sentences, the manuscript needs a thorough revision of English.
• Introduction and background have been narrated well; however, there is a scope and I have made some suggestions in the manuscript (already marked in the pdf file).
• The authors should address on the emerging pollutants [Polycyclic aromatic hydrocarbons (PAHs), presence of manufactured nanomaterials in packaged foods and their exposure to the environment, pharmaceuticals and personal care products, nano and micro plastic disposal [used in face mask, food pakaging materials etc.]. There is a chance to pollute the ecology of any tourist spot by the emerging pollutants.
• Structure conforms to journal standards; however, authors are requested either to provide data on emerging pollutants for a period or to add a separate subheading as ‘Future Scope of Research’ before conclusion.

Experimental design

• The study is based on the original primary research within scope of the journal.
• Research is relevant and appropriate.
• Materials & methods: Needs to be improved
• The article has potential in the present context.
• Methods described have some limitations that need to be clarified and described properly.
• While describing results, authors mentioned data collection and analysis part. It is to be kept in the M&M. There is a need for revision of results part and results are to be narrated precisely.

Validity of the findings

• The research has the novelty and authors addressed the same.
• Based on the manuscript authors have written the conclusion; however, before conclusion they should keep a subheading on the future scope of research highlighting emerging pollutants and their threats.

Additional comments

1. I have suggested to change the title.
2. Recast of the entire manuscript is needed by avoiding the repetition of research background.
3. The topic is highly relevant and such article can be considered after major revision.

Reviewer 3 ·

Basic reporting

• Heading number 2 (Research Area) should be included in Methodology part as it seems to be the part of material and method.
• Line 90: Should be rephrased like "Previous studies have shown that tourists in the scenic spot have a great impact on the physical and chemical properties [12-15]"
• Similarly, sentence of line 171-172 seems to be repetitive as 173-174. Should be rephased.
• Line 160-161: rewrite the sentence.
• Whole manuscript needs to be revisited for language improvement to ensure the better comprehension.
• As per the journal’s guidelines, Full mane of journal should be italic; may be followed same for each reference.
• Reference number 12; “Li Mei, Yang Wanqin, Xiao Yan, etc” should be “Li Mei, Yang Wanqin, Xiao Yan, et al.”, pursue for other references (17) as well.
• References are not in uniform style; uniform reference style should be followed as per journal’s guideline.

Experimental design

Expeimental design used in the experiment is clealy descibed with sufficient information. Besides, some of the improvements are needed like:
• Line 120: External heating K2Cr2O7 bulk density ratio method used for ecological SOC determination need to be discussed in detail and give reference.
• Line 136: cited authors “Din Yuanhao and Xie Deti” is not correctly cited. It can be written as “Yuanhao and Deti” if it’s a single paper. Cited publication is missing in references.
• SRI should be abbreviated wherever used first. Initials used (Sij, Soj,) in the formula for SRI should be abbreviated.

Validity of the findings

• Results represented for the respective analysis are starting with methodological descriptions which can avoided (4.1.1; 4.1.2.; 4.1.3.......)
• Title of figure-1 should be corrected as it is representing "pH" as "PH".
• Title of figure-2 can be revised as the data presented is not only enzymes but organic matter and nitrogen as well.
• The data pertaining the bulk density (g/cc) presented in Table-2 seems so high, cross check.
• As described in methodology section 3.4, correlation studies were carried out but not included in the result section. Can be included for better representation, if available.
• Statistical comparison is lagging in the result part which is essential for better understanding and data interpretation.
• Discussion part is adequate and nicely written however, it can be further strengthened by supporting the scientific findings and valuable references.

---

## Round 0.2 · Minor Revisions

Your manuscript is much improved from the previous version. However, Reviewer #3 has identified some minor issues that need to be addressed before your manuscript can be accepted for publication.

Reviewer 3 ·

Basic reporting

Although authors have improved the manuscript but there are some of the things which still need to be addressed.
In text citations: name of authors and year should not be italic. In text citation should be done using authors surname rather than initials. Need to be thoroughly checked.
Most of the references are not in proper formatting.
If the citations are intext (using author names), why shorten the reference numerically? These should be alphabetically shorted.
I think it’s not needed to highlight sections like objective, methods, conclusion etc. in the abstract.
What is the H response? How was it calculated? Need to be highlighted in the manuscript.
pH should be written as "pH" instead of PH.
The author should thoroughly pursue the manuscript to ensure the much needed qualitative improvement.
Discussion "section-4.1": Numberings in discussion paragraphs may be avoided.

Experimental design

NA

Validity of the findings

NA

---

## Round 0.3 · accepted · Accept

The authors have successfully addressed and implemented the revisions suggested by the previous reviewer and the manuscript can now be accepted for publication.